# Integrated Robotic and Network Simulation Method

**DOI:** 10.3390/s19204585

**Published:** 2019-10-21

**Authors:** Daniel Ramos, Luis Almeida, Ubirajara Moreno

**Affiliations:** 1Faculty of Electric Engineering, Federal University of Uberlandia, 38701-002 Patos de Minas, Minas Gerais, Brazil; 2CISTER, Instituto de Telecomunicações, FEUP-University of Porto, 4200-465 Porto, Portugal; lda@fe.up.pt; 3Automation and Systems Department, Federal University of Santa Catarina, 88040-900 Florianopolis, Santa Catarina, Brazil; ubirajara.f.moreno@ufsc.br

**Keywords:** networked robotic systems, robot cooperation, communication network simulation, simulation framework, simulation method

## Abstract

The increasing use of mobile cooperative robots in a variety of applications also implies an increasing research effort on cooperative strategies solutions, typically involving communications and control. For such research, simulation is a powerful tool to quickly test algorithms, allowing to do more exhaustive tests before implementation in a real application. However, the transition from an initial simulation environment to a real application may imply substantial rework if early implementation results do not match the ones obtained by simulation, meaning the simulation was not accurate enough. One way to improve accuracy is to incorporate network and control strategies in the same simulation and to use a systematic procedure to assess how different techniques perform. In this paper, we propose a set of procedures called Integrated Robotic and Network Simulation Method (IRoNS Method), which guide developers in building a simulation study for cooperative robots and communication networks applications. We exemplify the use of the improved methodology in a case-study of cooperative control comparison with and without message losses. This case is simulated with the OMNET++/INET framework, using a group of robots in a rendezvous task with topology control. The methodology led to more realistic simulations while improving the results presentation and analysis.

## 1. Introduction

The increasing use of teams of mobile cooperative robots in a variety of applications including area coverage, exploration and cooperative transport, is also pushing research on cooperative control solutions. In general, cooperative robots may be considered as a set of movable sensors that exchanges information to complete a task. Nevertheless, these solutions are built on top of a communication network that has several imperfections such as delays and packet losses, ending up having a significant impact on team behavior [1], especially on decentralized control cases that rely heavily on explicit communication between robots [2].

These solutions are developed, tested, and validated through simulations, emulations, testbeds, or real robots [3,4,5]. Although these four concepts are not mutually exclusive, choosing to use one or all of them depends on material and financial resources available, study deadlines, project knowledge and personal experience. A convenient combination is to carry out simulations first, for their flexibility, and then implement the real system. However, moving from simulation to implementation can be time and effort consuming, despite new model-based engineering approaches that try to automatize this transition, but which are still limited in their capabilities. The effort increases significantly when an unknown design problem is only identified later at testing time, forcing redesigns, new simulations and more implementation.

One approach to soften simulation to implementation transition is to improve simulation accuracy, aiming at results that are more realistic. This approach has been followed in different ways, by integrating different robot physical aspects in 3D simulations as in eRobotics [6], or integrating different levels of implementation abstraction as in meta-modelling simulation for robotics [7] and in Cyber-Physical Systems (CPS) [8] or yet, by integrating a communication network simulation with a desired application as in Networked Control Systems (NCS) [9], Wireless Sensor Network (WSN) [10], Internet of Robotic Things (IoRT) [11], and in teams of mobile robots [10,12,13,14].

Simulation of integrated application and network models has been achieved in diverse ways: co-simulating with two different simulators [12], expanding a network simulator with physical models [1], or by expanding the physical simulator with network models [9]. Even though the referred works represent advances in studying the impact of a network on robotic systems, they lack a systematic method to build and evaluate a complex simulation study.

Several such methods do exist, though, aiming at different science domains such as general model simulations [15], CPS [8], Digital Twins [16], Network Simulation Only [17], and Network Emulation Only [3]. In the domain of robotics, methods often describe procedures related to a robotic task, e.g., a method for using a simulation framework to study mobile robots operating on uneven terrain [4] and a method for abstracting expression techniques for diverse types of robots [18]. Other general methods for robotics focus on different robot aspects, e.g., combining physical domains as in eRobotics [6], or initial ideas that still need to be improved, as in IoRT [11].

In our domain of interest, namely networked control systems made of teams of cooperating robots, an initial concept method has been shown by the authors, as a work in progress, to improve the assessment of the network influence on cooperative strategies [1,19,20]. However, several improvements could still be made, finally resulting in what we denominated as the Integrated Robotics and Network Simulation (IRoNS) method, which incorporates known validation and documentation techniques. Particularly, we make use of an initial method concept presented in [20] that relies on using OMNeT++/INET for improved network simulation accuracy, and we extend it with a four-step validation [21] combined with confidence interval statistical analysis [22], a factorial experimental design with confidence interval analysis [22], and a communicative modelling process [23].

We demonstrate the proposed methodology in one detailed case-study that compares three rendezvous control strategies under a faulty communication network. The resulting method still maintains realistic simulations as initially shown in [1,19], but also improves the resulting documentation, presentation and analyzability of the results.

The paper is organized as follows. The next section summarizes related works. Section 3 describes the IRoNS method, detailing its critical points. Section 4 describes a simulation study using the method. Section 5 presents final considerations.

## 2. Related Work

The use of methods to guide simulation studies is a well-known topic for general system modelling. Works in this domain, as reviewed in [22], concentrate on providing lifecycle workflows and guides for best practices during simulation development. The main idea, common to these methods, is to set the study in several sequential stages consisting of: 1. Planning, problem modelling and documentation; 2. Simulation implementation; 3. Simulation validation; and 4. Experimentation and Analysis. The last two steps are subjects of Validation, Verification, and Test (VV&T) techniques [24], which are often used for establishing the credibility of the simulation study [15].

These works are suitable as a general guideline for common simulation studies, although they do not contemplate applications particularities, lack technical depth and are harder to apply when dealing with complex simulations. These characteristics motivated the appearance of new research lines to deal, for example, with agents in Agent-Based Modeling and Simulation (ABMS) [21,22,25]. ABMS appeared as a reaction to the lack of formal basis in simulation studies, incomplete documentation and missing results reproducibility [21,25]. However, as the concept of agent can appear in a wide selection of applications, related works tend to focus on common aspects but only on specific topics inside a simulation development lifecycle, e.g., choosing a validation technique for agent-based simulation [21], proposing a formal modelling process [25], or a formal documenting process [23].

Even though the above-mentioned methods, procedures and recommendations give important and useful insights about building a simulation study, there is another consideration that should be taken into account when working with an integrated simulation of a cooperative robots team and its communication network, i.e.,: it is a cross-domain simulation.

The work in eRobotics [6], a branch from eSystem Engineering, deals specifically with this type of situation. At first, eRobotics targets the development of methods and concepts, refining engineering processes and using semantic modeling techniques, to provide the necessary models for a robot 3D simulation. With this basis, a Virtual Testbed is built, consisting of a 3D simulation software environment for the integrated cross-domain development of complex systems based on 3D models.

In another work related to eRobotics methodology [16], the authors combine the concept of Virtual Testbeds with another approach from industry, namely the notion of Digital Twins, which consists of real world objects with a corresponding virtual representation capable of communication and acting as intelligent node in the Internet of Things. The authors use the resulting Experimentable Digital Twins as a core of the simulation-based development process, enabling detailed simulations at system level and realizing intelligent systems focused on 3D modeling of physical aspects of a robot.

These works are examples of methods used in robotics with the objective of improving robot simulation characteristics, but in a different scope with respect to the work proposed in this paper. Here, for a cooperative robotics simulation, the communication network is the object of interest and robots detailed physical properties assume lower relevance [13].

Cooperative robotics and network simulation received more attention in recent years as the development of network simulators advanced, enabling the implementation of complex interactions inside the network simulator [1] and enabling cross-simulator communication [12]. Researches into using this type of simulation are sparsely distributed along several lines, with the most recent ones concentrating on Networked Robotic Systems (NRS) [20] and Internet of Robotic Things (IoRT) [11].

Another approach to deal with cooperative robots and networks is to use hardware and software tools to enable assembling and studying swarms of general-purpose robotic systems [26]. The main idea is to deploy the solution in hardware and see how all the algorithms work together. This approach led to positive results as shown in [27] for swarm response and in [28] for topology dynamics, but it requires hardware implementation and can be exhausting when several testing cases are required. Our proposed approach differs from [26] because ours is simulation-based, addressing the case when there is a need of exhaustive simulations or when the hardware testbed is unavailable. Moreover, the use of both approaches can contribute to increase validity of the results.

In the case of simulations studies, in the same way as mentioned before for ABMS, they are currently lacking a formal basis for techniques and documentation procedures. This problem motivated our initial method concept [20], which already proved useful to assess simulations in [1,19]. The IRoNS Method that we propose now extends previous related works by incorporating proven validation and documentation techniques, making the simulation process more robust and useful, and with a better result presentation.

## 3. IRoNS Method

The proposed method uses a workflow structure to guide the study simulation development, dividing the study in three development stages: Problem definition, simulation framework, and experimentation. The last two stages can be further divided following a ‘plan, execute and assess’ methodology, which leads to a final 10-steps procedure that is briefly described below (Figure 1).
**Problem Formulation (1):** The first part of the method is to state and define what kind of problem is going to be the object of study. In general, this initial part derives from an initial study or a demand that urges for a complex simulation for validation.**Choosing Solutions (2):** After defining the problem, it is necessary to choose the techniques that allow solving the problem, i.e., techniques that will be simulated to emulate and assess the situation referred in the problem study. It is important to note that the IRoNS Method does not depend on any specific robotic or network techniques.**System—Specifying (Initial Documentation) (3):** Consists in organizing all the information decided so far, describing the concepts behind the chosen techniques and giving a special attention to creating an assumptions document, which should be updated during the entire development cycle.**Base Simulation—Planning (4):** The next step consists in planning the simulation framework (Base Simulation—BS), which is also referred to as conceptual and communicative modelling. This modelling includes documentation procedures with different objectives: firstly as a visual documentation to guide simulation implementation and secondly as textual documentation for study reproducibility. The idea is to gather all necessary information about the simulation study and framework before starting the implementation, including the system structure, expected interactions, simulator selection, desirable simulation characteristics, initial parameters values, and any other related information. The main documented topics are described in Section 3.1.**Base Simulation—Implementing (5):** after the documentation, the focus now is implementing the simulation. In this case, we opted to continue extending a network simulator simulation with cooperative robotics features, as shown in [20]. The OMNeT++/INET [29] network simulator was selected for its modularity, graphical interface, community support, and easy code debug. Moreover, we did not observe significant performance differences between network simulators that could justify using another simulator.**Base Simulation—Validating (6):** The resulting simulation, as referred a priori, is a candidate base simulation until it passes through a validation process, which verifies whether it keeps the main characteristics of a real system or of the original simulation [30]. This part uses Verification, Validation, and Test (VV&T) techniques, integrated with statistical analysis with confidence interval, which is further detailed in Section 3.2.**Case-Study—Experimentation Design (Planning) (7):** Once the candidate simulation passes the validation process, it becomes a Base Simulation that is ready for experimentation. However, it is necessary to plan case studies to make sure they are aligned with the objectives defined in the first step of the method, thus requiring an experimentation design. A suggestion of experimental design is presented in Section 3.3.**Case-Study—Experimenting (8):** This step consists in implementing the planned studies, executing simulations and gathering experimental results. The main concern, here, is to enforce experimental rigor to avoid collecting incorrect data or producing incorrect behaviors.**Case-Study—Data Analysis (9):** Once data is collected, it must be analyzed and converted from raw into useful information, also presenting it in an adequate form. Structuring these results as defined in the experimentation design allows using statistical analysis with confidence intervals to assess their significance. Detailed execution of this item for this type of simulation presented in Section 3.4 is the main contribution of this paper as the use of confidence intervals integrated with experimental design contribute to better results presentation and validation.**Conclusion (10):** The last step is to check if the obtained results are enough to satisfy the study objectives, answering the problem study. If results are deemed not good enough, the method cycle should be iterated.

The main objective of this method is to provide a systematic tool to develop a simulation study in sequential steps, considering the particularities of this type of simulation, i.e., the cross-domain complexity. Our main contribution lies in integrating techniques in four critical points, namely for simulation planning, base simulation validation, case-study experimentation, and result analysis, which we describe in the following subsections. An example use-case is presented in the next section.

### 3.1. Base Simulation—Planning

Due to its impact in the entire process, we consider this step a critical point, consisting in preparing information to simplify simulation implementation and improve simulation reproducibility, a step also known as conceptual and communicative modelling [22].

This type of modelling is usually simple and easy to use, but its complexity may grow quickly in the following three situations: (1) If there are several developers working in the same implementation; (2) when simulation is complex and involves several views; and (3) in the case of poor team knowledge on the simulation topics. For integrated robotics and communication simulations, all three cases may be true, requiring more details and explanations in the conceptual model. In practice, this implies a large and detailed documentation process that, yet, can be too complex to be used as guide in implementation or too simple to help others understand the simulation topics.

A common recommendation is to maintain conceptual modelling simple and define a communicative model that is more refined, with more detailed information. Therefore, the computational model is not the main concern in this part and the generalized idea of how simulation will work and its functionalities are the key points [31]. An extensive source of information of this modelling is presented in [32]. We opt to adapt and use the recommendations of a simpler conceptual modelling described in [25,31], which uses diagrams, figures, conceptual maps and other visual forms to describe the simulation.

On the other hand, communicative modelling focuses on documentation, in a textual form, being one of the most important aspects of the simulation, aiming at a better understanding of the simulation by others. For this modelling, we adopt the ODD protocol [23], modifying some of its terminology and adding particularities of the type of simulation we are addressing. The ODD protocol uses the following documentation topics:
**Purpose:** Consists of the same initial documentation already made within the IRoNS method, indicating the simulation main motivation and what to expect from it.**Entities, state variables, and scales:** Consists in defining what is relevant for the study in terms of algorithms, evaluation parameters, observation parameters, measurement units and, specifically in this context, robot and cooperation characteristics.**Process overview and scheduling:** Defines how algorithms are organized, what they do and in which order. In our context, it is especially important to define relationships between the network, topology control and the robots cooperative control.**Design concepts:** There are eleven design concepts in the ODD protocol [23] describing the application of an agent simulation. Using these concepts for robot simulation is straightforward if we consider the robot as a particular physical agent and the set of cooperative robots as a physical multi-agent simulation. All information regarding the simulation of robots, cooperative control and cooperation is documented here.**Initialization:** Describes the simulation initial conditions, initial values and the expected effects on the concrete simulation case.**Input data:** This topic is needed when using input data from external sources or another simulation software, only.**Submodels:** Description of each submodel used in the simulation. Here we include all the network and topology control aspects that were not described before. Any details about extra modules and the simulation environment must also be included here.

### 3.2. Base Simulation—Validation

After finishing the first simulation implementation, referred as a candidate Base Simulation, the simulation needs to pass a validation process. As there are diverse types of algorithms, it is not trivial validating them all at the same time, and it would probably require a testbed [33] or real robots to reproduce real-world results for validation, which may not be available.

One approach is to consider individual domain validations and assume composability, i.e., the individual validations will remain valid upon integration. Thus, the resulting simulation should give some indications of the overall behavior of the system and possible algorithms interactions, even if the results are not precisely accurate.

There are several techniques already developed for simulation validation [30,33], but we adopt a combination of a simple four-step ABMS validation technique [21] with a statistical analysis via confidence intervals (CI) from VV&T research [22].

The four steps in [21] consist of face validation, sensitivity analysis, calibration, and statistical validation. Face validation consists in a human expert analyzing the overall behavior of the system, making the necessary modifications to achieve expected results or to eliminate visibly bad results. Sensitivity analysis aims at assessing the impact that parameter changes have on the simulation behavior. The third step is the calibration, adjusting parameters to increase model accuracy and reevaluate its final values.

The final step is the statistical validation, for which we use a confidence interval technique using the errors between multiple simulation instances [22]. For this purpose, we formalize the technique as follows: Let *R*_1_ and *R*_2_ be the statistical results from the same situation simulated on different simulation frameworks, Simulator 1 and Simulator 2 respectively, *Z_j_* = *R*_1*j*_ − *R*_2*j*_ as the error achieved in an experiment *j*. The mean $\bar{Z}\left(n\right)$ and variance $\hat{Var}\left[\bar{Z}\left(n\right)\right]$ are then computed for a given number *n* of experiments (samples).

The confidence interval (*CI_z_*) is defined by (1) using $\bar{Z}\left(n\right)$ with $\hat{Var}\left[\bar{Z}\left(n\right)\right]$ used to determine the half-length (*L_z_*) of the interval (2) in terms of $t_{n-1,1-\frac{\alpha }{2}}$ critical points of a *t-student* distribution, obtained from a *t-student* table [22] as function of *n* − 1 samples and an auxiliary constant α, determined by the desirable confidence C*_d_* (3).
(1)$$CI_{Z}=\left[\bar{Z}\left(n\right)+L_{Z}\;\;\;,\;\;\bar{Z}\left(n\right)-L_{Z}\right]$$
(2)$$L_{Z}=t_{n-1,1-\frac{\alpha }{2}}\sqrt{\hat{Var}\left[\bar{Z}\left(n\right)\right]}$$
(3)$$\alpha =2\cdot \left(1-\frac{C_{d}}{100}\right)$$

The resulting interval can be assessed in two ways: statistical and scale significances. If the confidence interval contains zero, it means that, with a determined confidence, the true mean of the difference between both implementations (the error) can be zero, thus, it is not statistically significant. However, when the confidence interval is too wide (low precision) or has low values when comparing with the overall result, the difference can be insignificant even if the interval does not contain zero.

### 3.3. Case-Study—Experimentation Design

Once the Base Simulation passes the validation process, we assume that it is a valid simulation model and it can be used for designing the experimentation, which is a process of planning simulation cases to analyze the impact of changes in parameters and algorithms on simulation results.

The initial goal is to define evaluation and observation parameters. Observation parameters are those used to track system states. Evaluation parameters are simulation performance criteria, which generally depend on the nature of the cooperative task that we are analyzing, and it can also consist of a combination of several observation parameters.

To do the case-study planning, we use a *factorial 2K experimental design*, where *K* is the number of parameters used. This approach measures the impact on the results when changing one parameter value or algorithm and can be combined with statistical analysis by confidence intervals when assessing the results. The *2K factorial design* has an increasingly demanding cost as the number of parameters and experiments increases, but it is very straightforward when these numbers can be bounded to a small size [22].

In this formulation, we must choose two levels (values) for a parameter or use two test algorithms, adopting ‘−’ and ‘+’ for their representation. In this case, by convention, ‘−’ is used for the parameter standard value or standard algorithm, i.e., the ones used in Base Simulation, and ‘+’ to indicate the test value or test algorithm that we are comparing with.

These results can be organized in an experimentation table, as indicated in Table 1 for an example with two factors (*K* = 2, indexed by *f* = 1…*K*). In this case, there are four possible factor combinations, indexed by *k* = 1…2*^K^*, and their respective results R^k^_j_. Moreover, we should explore *j* = 1…*n* different simulation situations (samples), under the same operational parameters, leading to *n* experimental tables and, in this case, $n\times 2^{K}$ simulations. For the cooperative robotic tasks addressed in this work, as an example, this can be achieved using different values for robots’ initial positions and by varying their initial communication topology.

The number of samples has a key role when building the statistical analysis by confidence interval, thus it is recommended to simulate at least 15 different samples to obtain satisfactory results using t-student distribution and 30 samples when a normal distribution behavior is desired [22]. A higher number of samples may improve results, as it may reduce the confidence interval length, however it can be time consuming and not always possible.

### 3.4. Data Analysis

After having the case-study planned and simulated, the last critical point is to assess the results. This assessment consists in determining how much each change, in a parameter value or in a technique, affected the task results, which is referred as ‘factor impact’. It is also possible to make this analysis to determine how much these changes interact with each other, which in this case is referred to as ‘cross-factor impact’.

In other words, to determine the influence of each factor on the simulation results, we compute each factor impact and the cross-factor impact. The factor impact e^f^ measures how much impact a change from state ‘−’ to ‘+’, in a specific factor combination *k*, has on results. The cross-factor impact e^fafb^ indicates how much two or more factors results are dependent of each other values.

Each impact value is obtained by using data from the experimentation table (Table 1). The symbols ‘−’ and ‘+’ are taken as corresponding scalar numbers ‘−1′ and ‘+1′ that multiply the respective result R^k^_j_ for each factor. To determine the factor impact, i.e., the average impact from changing values, we sum algebraically the results and divide by 2^K−1^ [22]. The cross-factor impact follows the same logic, just crossing the factors, as the name says. For the two factors example in Table 1, the resulting impact values are given by (4) for a sample *j*.
(4)$$e_{j}^{1}=\frac{-R_{j}^{1}+R_{j}^{2}-R_{j}^{3}+R_{j}^{4}}{2^{2-1}}\;e_{j}^{2}=\frac{-R_{j}^{1}-R_{j}^{2}+R_{j}^{3}+R_{j}^{4}}{2^{2-1}}\;e_{j}^{12}=\frac{R_{j}^{1}-R_{j}^{2}-R_{j}^{3}+R_{j}^{4}}{2^{2-1}}$$

These impact values can be used in the same statistical formulation mentioned in the validation process (1) to (3), thus obtaining the mean ${\bar{e}}^{\;f}\left(n\right)$ and variance $\hat{Var}\left[{\bar{e}}^{\;f}\left(n\right)\right]$ for *j* = 1…*n* experimental samples, and building the confidence interval $CI_{e^{f}}$ for each factor *f* impact, built from a half-length of $L_{e^{K}}$.

The resulting confidence intervals interpretation for each factor is similar as made in the validation. If the interval includes zero, the change in the related factor value did not impact on simulation results. However, if the interval does not contain zero, we can state that the change in this parameter had a significant impact on the results. Besides these aspects, we can also observe the cross-factor impact, which indicates if there is any significant interaction between simultaneous factor changes.

## 4. A Case-Study Illustrating the Use of the IRoNS Method

To show how the new method could improve results analysis with a set of algorithms of different types, we present an example case-study of cooperative strategy comparison in which we follow the IRoNS Method. The main idea is to show how these algorithms can interact and how we can analyze them before using a hardware setup.

### 4.1. Problem Formulation

The method workflow starts from a stated problem, which, in this context, involves cooperative robots and communication network topics. We define the comparison of control strategies as the target, including an assessment of their performance under network faults.

This problem needs to be further specified, as stated in the method first step. It is possible to choose any combination of algorithms and parameters. In this case, for simplicity sake, we chose to work with a limited group of 10 terrestrial and homogeneous mobile robots, executing a decentralized rendezvous task. These conditions were determined by convenience and do not represent a limitation of the framework. For example, considering more and heterogeneous robots can be accommodated at the cost of increasing the simulation complexity and thus the time needed for its execution.

In cooperative robotics, the rendezvous is a cooperative task where robots must agree on a meeting point and reach it without breaking communication links. This is a trivial problem in centralized cooperation, however it may become challenging as a consensus problem with decentralized cooperation, in which robots can only receive information through their direct communication neighborhood [34].

From the communication network side, a decentralized solution is desired, too, given its flexibility with respect to topology and resilience to network faults, which is particularly relevant for wireless communication with a simple communication topology management. The network model is based on OMNET++/INET IEEE802.11 (Wi-Fi). Other protocols can be used if available in the network simulation framework. For example, OMNET++/INET also supports Zigbee [35] and LORA [36].

### 4.2. Choosing Solutions

After the problem formulation, the next step of the method is to choose techniques that can solve the problem or can produce desired study circumstances. For this example, we will consider three control strategies for the rendezvous problem, the network protocol involved, and the topology control used, which are detailed bellow. The selection of these techniques considered the fact that, to the best of the authors’ knowledge, they were never simulated together nor compared in the scope of a simple robotics cooperative task. The objective here is to detect behaviors and interactions that do not show in a simple simulation that does not consider the network idiosyncrasies.

#### 4.2.1. Average Rendezvous

This is one of the simplest decentralized rendezvous techniques in the literature [37], in which the reference point for each robot is determined as the average value of its neighbor’s positions. At each time step *k*, each robot uses the received information to drive toward the updated average point (5), where *X_iref_* is the position vector reference for robot *i*, *X_j_* is the positions vector received from neighbor robot *j*, a*_ij_* is adjacency matrix element that indicates whether there is a communication link between robots *i* and *j*, and *j* is an index of *robot i*’s *n_i_* neighbors.
(5)$$X_{i_{ref}}\left(k+1\right)=\frac{\sum_{j=1}^{n_{i}}X_{j}\left(k\right)\cdot a_{ij}}{n_{i}}$$

#### 4.2.2. Circumcenter Rendezvous

This technique consists in defining, at each instant *k* and in each robot *i*, the smallest enclosing circle that includes all robot direct neighbors and uses its center as the robot reference meeting point [38] (Figure 2).

#### 4.2.3. MPC Rendezvous

The MPC (model prediction control) rendezvous [34] is an algorithm that uses a consensus formulation with receding horizon stated as a quadratic optimization problem. The optimization problem in robot *i* can be stated as finding the minimum of an objective function *J_i_* constrained by saturation velocities (6) [34].

The objective function is composed of a consensus function *f_i_* for robot *i* and its neighbors, a 2D velocity vector *V_i_* and auxiliary matrix *H_i_*. This auxiliary matrix includes the adjacent matrix elements and a ponderation factor that penalizes changes in control.
(6)$$\mathrm{minimize}\;\left(J_{i}=\frac{1}{2}\left[\begin{array}{cc} V_{i}^{x} & V_{i}^{y} \end{array}\right]\left[\begin{array}{cc} H_{i} & 0\\ 0 & H_{i} \end{array}\right]\left[\begin{array}{c} V_{i}^{x}\\ V_{i}^{y} \end{array}\right]+\left[\begin{array}{cc} f_{i}^{x} & f_{i}^{y} \end{array}\right]\left[\begin{array}{c} V_{i}^{x}\\ V_{i}^{y} \end{array}\right]\;\right)\;s.t.\;v_{\max b}^{x}\leq v_{i}^{x}\leq v_{\max f}^{x}\;v_{\max b}^{y}\leq v_{i}^{y}\leq v_{\max f}^{y}$$

This technique is considerably more complex than the rest and it was selected to show that even demanding algorithms that rely on optimization could be utilized in this type of complex simulation.

#### 4.2.4. Mobile Ad-Hoc Network

A MANET is a wireless mobile local area network that does not rely on a central point to coordinate message exchanges in the network, the nodes forward packets to and from each other on their own. We chose the IEEE 802.11b standard for this simulation, given its robustness, which is part of the IEEE 802.11 series of WLAN standards. Devices using IEEE 802.11b experience interference from other devices operating in the 2.4 GHz band, be it other IEEE 802.11 devices not engaged in the team or Bluetooth devices, microwave ovens, and cordless telephones. These devices can be thought of as alien uncontrollable network traffic generators, creating occasional collisions with the team transmissions and consequent message delays and losses.

This wireless network model was selected as a worst-case scenario and with the purpose of inserting complexity in the network part of the simulation. The network may be considered ideal or with any other type of technology or protocol.

We also use an overlay protocol on top of IEEE 802.11b that organizes communications in periodic rounds, i.e., communication cycles, within which the robots in the team transmit one at a time, evenly spaced, such as proposed in [2]. As in this type of simulation we have total control of the network, we opted for this protocol to demonstrate that additional adaptations and behaviors can be considered, adding further complexity to the simulation. We used a communication cycle of 100 ms, which we deem adequate for the dynamics of the robots.

#### 4.2.5. Topology Control

When dealing with wireless communication with limited range and with nodes that are constantly moving, the physical communication topology is naturally variable. However, it is possible to establish a logical communication topology, which goes beyond the physical topology, as it may ignore communication links or create virtual links through message routing protocols. This communication link management is called topology control.

The communication topology has a significant impact on cooperative behavior [28], not only in network properties, but also on the dynamics of decentralized control algorithms. Each robot uses information from its *n* neighbors and, as we increase the number of neighbors, it has more information to deal with. We may then reach a point in which the algorithm receives too many messages, generating too much information that does not contribute to the task, but increases overhead and may negatively influence the network performance. On other side, limiting the number of neighbors, e.g., using a fixed topology, reduces traffic in the network, which can be helpful in low bandwidth protocols. However, decreasing it too much will negatively affect the cooperative task.

In this example, we use topology control to provide a fixed logical topology during the entire simulation. This means that new communication links will not be created despite the robots approaching each other in the course of the rendezvous task. We have already shown that topology changes strongly impact the results when we compared fixed and dynamic topologies [19]. Thus, here we simply show that different numbers of fixed links already have significant impact on the task performance.

### 4.3. System Specification

Concerning documentation, beyond the documentation of the problem itself, each technique should be detailed with pseudo codes, possible interactions, important parameters and any other valuable information.

In what concerns the assumptions, since this study has the objective of verifying the impact of a specific set of techniques on the results, it is necessary to avoid any other influences and variables in the simulation. This leads to the following set of assumptions: robots are finite points in space, physical collisions are not considered, the environment is an open space without objects, the robot motion is determined by first order dynamics without uncertainty and each robot knows its own position with precision.

### 4.4. Base Simulation—Planning

The next step in the method is the simulation planning through conceptual and communicative modelling. However, for the sake of conciseness of this paper, we will not detail this step here. In fact, the conceptual model took 5 pages of figures and diagrams and the communicative modelling took 10 pages of detailed information.

The depth of the modelling may vary and will typically depend on the number of people involved in the design process and simulation study. It is important to ensure the entire design team has the same goals and simulation algorithms knowledge, thus the larger the team, the more detailed information is need. Nevertheless, even for small teams a reasonably detailed documentation is important for simulation reproducibility.

### 4.5. Base Simulation—Implementation and Validation

The base simulation was implemented in OMNeT++/INET [29] using 4.6 and 2.5 versions, respectively. We also use an auxiliary library for quadratic optimization, called Quadprog++ 1.2.1 to run the rendezvous algorithm presented in Section 4.2.3.

We applied the validation process to the control strategies referred above, which were originally simulated in Matlab. For this validation, we chose two evaluation criteria: the error in the rendezvous convergence time (*T_f_*) and the error in the convergence distance (*D_f_*). We used the same simulation conditions on Matlab and OMNeT++ and we built confidence intervals of the error between the results achieved with both simulators.

We followed the referred 4-step verification process, with 99% confidence intervals (CI) for the three control strategies, both assessment criteria and 15 samples, as indicated at Table 2. All CIs have relatively small length, indicating a good precision, and include zero, meaning that the error true mean may be zero. Thus the differences between both simulation frameworks are not significant, as we referred before, which is in agreement with our observations in [20] and validates our OMNeT++ framework.

### 4.6. Study Case—Experimentation Design

The experimentation objective is to compare the performance among the referred three rendezvous algorithms in terms of how long it takes for each one to conclude the task, with and without message losses. Fifteen initial conditions were sorted and applied in the same way to each algorithm, resulting in 15 samples of convergence time.

We used the following notation for the rendezvous algorithms: A—Average (Section 4.2.1), C—Circumcenter (Section 4.2.2) and MPC—Model Prediction Control (4.2.3). For the network parameter, we used: I—ideal network without losses and C—network with message collisions/losses.

Initially, for the sake of simplification, we compare algorithms in pairs, defining three comparison experiments: A × MPC, A × C and MPC × C. Formulating the experimentation we consider that the first algorithm of each pair assumes the ‘−’ factor role and the second the ‘+’ role. Thus, in the A × MPC case, for example, we want to analyze the performance gain/loss when changing from the Average to MPC technique.

### 4.7. Study Case—Experimentation and Data Analysis

After making any needed adjustments in the Base Simulation, we defined 15 different initial conditions (samples) and obtained results each factor combination. For each sample, factor combination results are transformed in factor and cross-factor impact.

The last step is to compute the mean and variance of the 15 samples of each impact factor and cross-factor to finally build their confidence intervals (CI). The CIs were built with confidence of 99% using (1), with α = 0.01 and t_14,0.995_ = 2.977, leading to the three CIs shown in Figure 3.

Figure 3 shows what happens to the rendezvous completion time when changing from the first technique X to a second technique Y (‘X × Y’), without any message losses and using a fixed topology control. In the first and last case, A × MPC and MPC × C respectively, confidence intervals do not include zero, indicating a significant performance difference. In particular, the convergence time increases when using MPC over an average rendezvous and decreases when using the circumcenter algorithm instead of the MPC. The confidence interval of the A × C experiment includes zero, meaning that these techniques do not have a statistically significant difference, which is consistent with the other two comparisons (A × MPC and MPC × C).

On the other hand, even the difference in convergence time between A or C and MPC is small, just around 1s, which may be considered negligible when compared with the absolute convergence times that varied between 20 and 50 s. With this information, we can conclude that changing between these rendezvous techniques does not impact significantly the convergence time under these network parameters, which is consistent with the Matlab simulation, too.

In the second part of the experiments we considered the robots deployed in an environment with network interference affecting the quality of robots communication through message losses. To create this effect in the simulation framework, we added a fixed node in the simulation environment that generates bursts of messages with a periodicity of 300 ms and duration of 100 ms, causing collisions with the robots messages.

We used the same 15 initial conditions of the previous experiment to build another comparison study between the first experiment, labelled ‘Ideal’ case, and the results achieved under message collisions, labelled ‘Collision’ case. Moreover, we considered that each robot can hold a message for 200 ms (one communication cycle memory), thus tolerating one message loss. However, if two or more consecutive messages are lost, the communication link is disabled until the robot receives a new message from that neighbor (source).

The resulting CIs between control strategies in a network with message losses are shown in Figure 4 and the comparison for each technique between the cases of without and with losses is shown in Figure 5.

These results show an interesting behavior change when comparing the rendezvous techniques with (Figure 4) and without (Figure 3) message losses. Now MPC takes significantly less time to converge than A or C, as the resulting CIs from A × MPC and MPC × C indicate. Again, the difference between A and C is not statistically significant, which is also consistent with the other two comparisons.

When comparing each technique without and with message losses (Ideal – I × Collision – C) (Figure 5), we see an increase of the convergence time for the three rendezvous techniques, being much stronger for A and C, indicating that these techniques are more sensitive to message losses. This explains the change in behavior observed when comparing the three techniques with and without message losses (Figure 3 versus Figure 4). It is important to note that these results could not be obtained with a simple robot control simulation in Matlab.

Given the observed impact of message losses, we carry out another comparison to answer the following question: What is the impact of, upon losses, keeping using the information of the last received message for a longer time (more communication cycles), thus tolerating more consecutive losses?

Another set of simulations were carried out using similar parameters as before, except for an extended memory capacity, or in other words, when messages are lost, the last message information is used for up to two communication cycles, i.e., messages were kept for up to 300 ms. The results are shown in Figure 6 and Figure 7.

When comparing results of a longer information lifetime, from one cycle memory in Figure 4 to two cycles memory in Figure 6, we observe that an increased lifetime reduces the differences in convergence times to values closer to the case without message losses (Figure 3), meaning that the different techniques exhibit closer behaviors. Moreover, the relative behavior among the three rendezvous techniques with increased information lifetime is still the same in the two cases with message losses, i.e., MPC takes less time to converge. However, there is now a more significant difference between A and C, in favor of the Circumcenter (C) technique, and consequently a less significant difference between MPC and C (note that the CI now includes zero).

This reduction in the impact of message losses is observed in Figure 7 too, when we compare again each technique without and with message losses but now with longer information lifetime. Both average and circumcenter rendezvous techniques show a moderate increase in convergence time, slightly more pronounced for A, while the impact on the MPC technique is not statistically significant (CI includes zero).

To further show how the method can be modular and how these parameters can affect the results, we made a simulation case in which we increase the number of robots and use the change in the topology (Linear × Grid, as indicated in Figure 8) as factor *e*1 and the change in control algorithm (Average × Circumcenter) as factor *e*2.

The results are presented in Figure 9, where we can clearly observe that the increasing number of robots also increases the convergence time.

There is a notable difference between the linear and grid topologies results. The first one heavily impacts the results as we include more robots, although in the second case, the variable number of robots is mitigated as we maintain 2 to 4 communication links between them. Moreover, we can also infer that both algorithms have similar performance under the grid topology, but under linear topology the circumcenter algorithm has a better result.

Using confidence intervals as proposed in this work (Figure 10), we can obtain the same information from Figure 8, where the first factor indicates that there is a significant reduction in the convergence time when changing from linear topology to grid topology. As the difference between both cases is constantly increasing as we increase the number of robots, the CI is larger than in the previous cases.

Concerning the second factor, we can observe a difference between circumcenter and average rendezvous, which is significant for the linear topology but minimal for the grid topology. This information is also obtained from analyzing the cross-factor CI “*e*12”, which shows that results interpretation needs to consider each factor value. In other words, the convergence time in this study varies with the chosen topology but also with the control algorithm. We can also understand that if we want more information about one of these factors, we need to select a fixed value of the second one for the simulation study.

## 5. Discussion and Conclusions

In this work, we discussed the process of building an integrated simulation case for networked cooperative robots. We proposed a novel method, the IRoNS Method, featured with documentation procedures and statistical analysis.

To illustrate the use of the IRoNS Method, we presented a case-study consisting on a comparison of diverse control strategies for a rendezvous task, under the influence of message losses. We showed how this method supports an objective comparison among the techniques under analysis in different operational scenarios, highlighting their behaviors and testing alternatives to improve them. In this case-study, we observed a change in the results of the techniques when comparing a fault-free with a faulty network scenario and assessed the effectiveness of a possible method to mitigate such faults.

Based on this example study, we can state that the three considered rendezvous techniques have similar performance in convergence time as long as message exchange is reliable. However, the MPC technique showed to be less sensitive to message losses, performing better under such network conditions. Moreover, we observed that the impact of message losses can also be mitigated by increasing information lifetime, i.e., maintaining previous information whenever messages are lost.

Obtaining this kind of information in a simulation phase was only possible with the integrated simulation framework offered by the IRoNS Method that provided a quick comparison and assessment using the confidence intervals statistical technique. Having this information in a simulation phase, developers can decide if any of these behaviors is acceptable or preferable before starting a real robots team implementation. Needed modifications can be done at simulation time, thus reducing transitions between simulation to real implementations and consequently, reducing the project total time.

Moreover, the IRoNS method has proved helpful in simulation planning, organization, implementation, and analysis, improving the accuracy of comparisons and helping the transition to the implementation on real robots. Its use is not bound to any specific operational or simulation framework. The results analysis can be uniformly made for any simulation parameter change, requiring only to establish a task performance metric.

We are currently applying this method to carry out comparisons among more complex cooperative tasks, such as advanced topology controls, using multiple operational scenarios and control strategies.

## Figures and Tables

**Figure 1 sensors-19-04585-f001:**
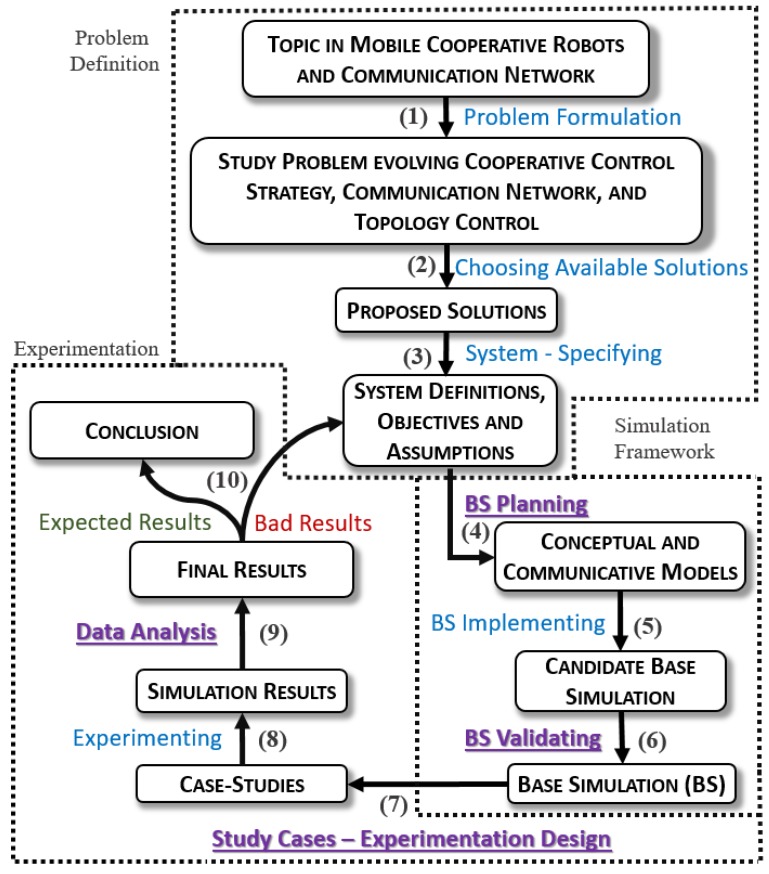
The Integrated Robotics and Network Simulation (IRoNS) method to build integrated robotics and network simulation studies, represented in terms of actions (arrows) and results (boxes). The underlined actions indicate critical points detailed in this paper.

**Figure 2 sensors-19-04585-f002:**
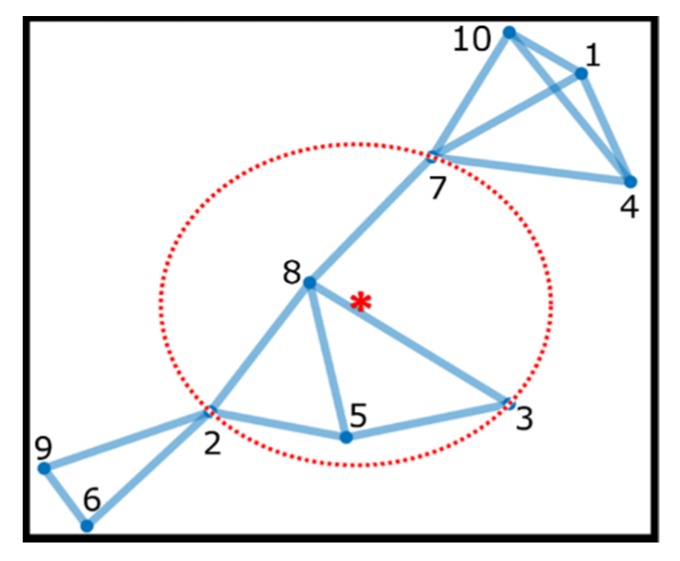
Example of circumcenter rendezvous showing the smallest involving circle around robot 8 direct neighbors and its center as reference for that robot in that time instant.

**Figure 3 sensors-19-04585-f003:**
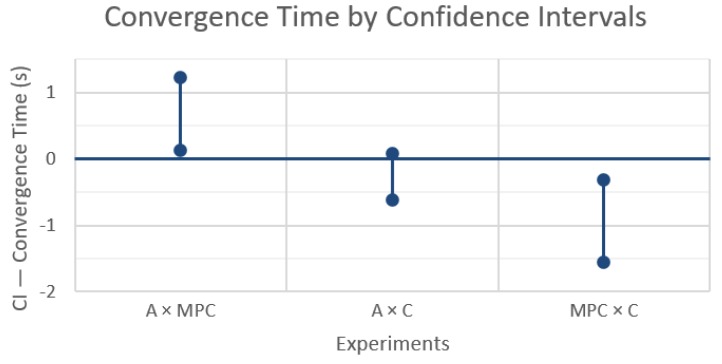
Differences in convergence time between the rendezvous techniques with 99% confidence intervals.

**Figure 4 sensors-19-04585-f004:**
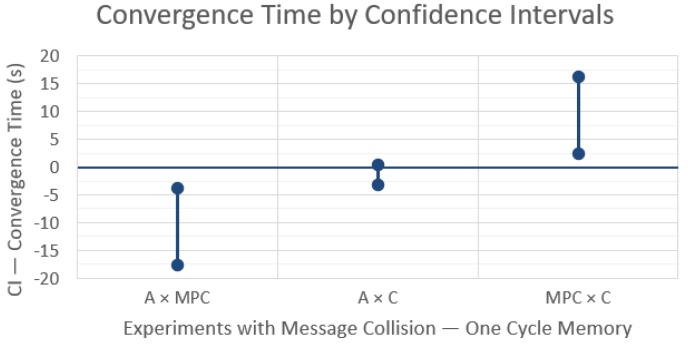
Differences in convergence time between the rendezvous techniques with 99% confidence intervals and message losses, with one communication cycle memory.

**Figure 5 sensors-19-04585-f005:**
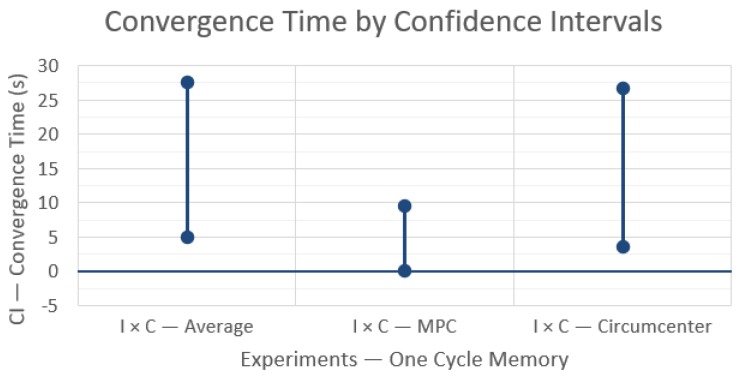
Differences in convergence time for each rendezvous technique with and without message losses, with one communication cycle memory, and with 99% confidence intervals.

**Figure 6 sensors-19-04585-f006:**
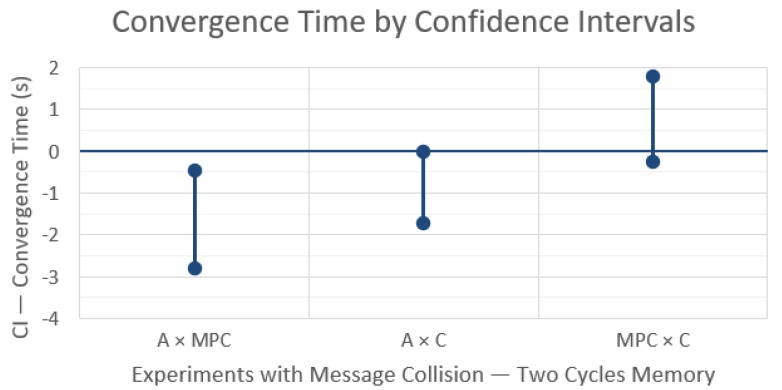
Differences in convergence time between the rendezvous techniques with 99% confidence intervals and message losses, with two communication cycles memory.

**Figure 7 sensors-19-04585-f007:**
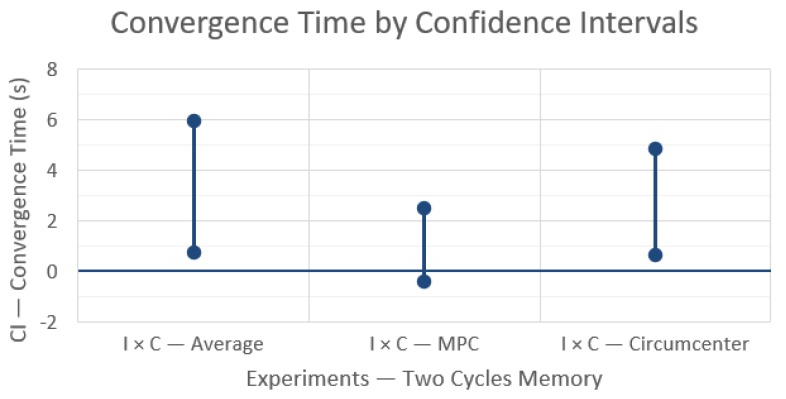
Differences in convergence time for each rendezvous technique with and without message losses, with two communication cycles memory, and with 99% confidence intervals.

**Figure 8 sensors-19-04585-f008:**
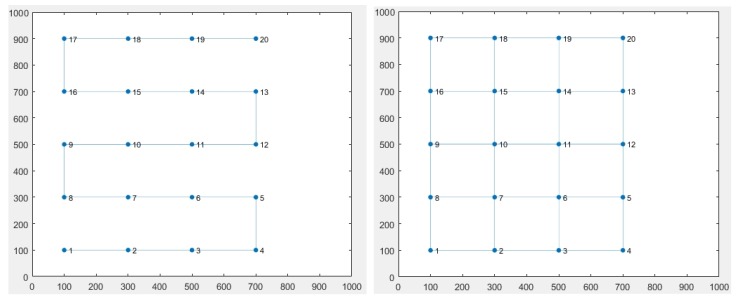
Fixed communication topologies for 20 robots.

**Figure 9 sensors-19-04585-f009:**
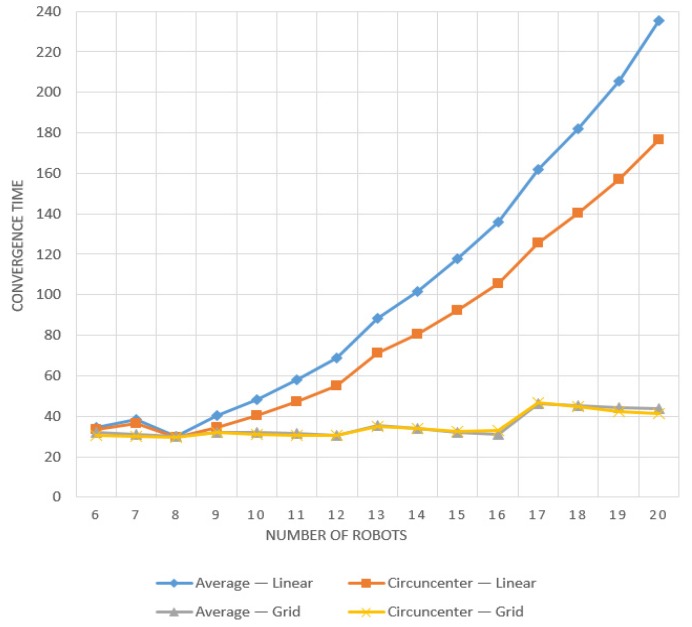
Results of varying the number of robots with two different control algorithms (Average × Circumcenter) and two fixed communication topologies (Linear × Grid).

**Figure 10 sensors-19-04585-f010:**
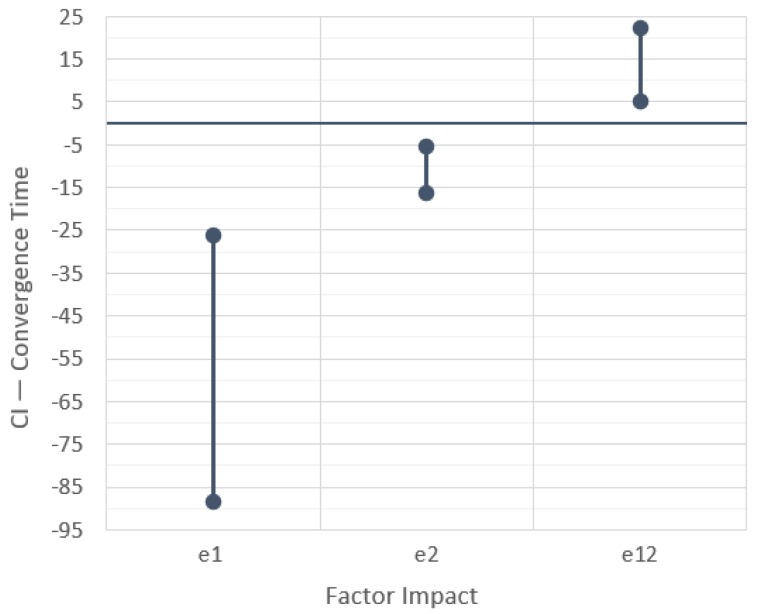
Differences in convergence time when varying the number of robots with two different control algorithms *e*1 (average × circumcenter) and two fixed communication topologies *e*2 (linear × grid), and its cross-factor relationship, with 99% confidence intervals.

**Table 1 sensors-19-04585-t001:** Factorial experimental design with two factors (K = 2).

Factor Combination (k)	Factor 1 (f = 1)	Factor 2 (f = 2)	Result (R^k^_j_)
1	−	−	R^1^j
2	+	−	R^2^j
3	−	+	R^3^j
4	+	+	R^4^j

**Table 2 sensors-19-04585-t002:** Base simulation validation with confidence intervals.

	Fixed Topology
**MPC—CI 99%—Tf**	[−0.016; 0.041]
**MPC—CI 99%—Df**	[−0.013; 0.0298]
**Average—CI 99%—Tf**	[−0.01; 0.021]
**Average—CI 99%—Df**	[−0.013; 0.022]
**Circumcenter—CI 99%—Tf**	[−0.021; 0.016]
**Circumcenter—CI 99%—Df**	[−0.023; 0.018]

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
