# Peer review of "Integrated Robotic and Network Simulation Method"

_sensors, 2019, doi:10.3390/s19204585_

Round 1

Reviewer 1 Report

When developing and deploying applications involving cooperative robots, there is a huge research effort on cooperative strategies involving communications and control. Simulation is generally used to first quickly test the proposed algorithms so that exhaustive tests can be conducted before the system is implemented in a real application. One major problem of this strategy is that in situations where the results of the implemented system do not match those of the simulated system, substantial amount of effort is required to rework the system.

To address this problem, the authors present a set of procedures known as Integrated Robotic and Network Simulation Method (IRoNS Method) for guiding developers in building simulation studies for cooperative robots and communication network applications. Their proposed solution exemplifies the use of improved methodology in a case study of cooperative robotic control with and without message losses. The authors used OMNET++/INET framework, using a group of robots in a rendezvous task with topology control.

Reading through this article, I would say that it's been well and comprehensively written. The presentation of the proposed solution is excellent and the main contribution of the proposed solution has been clearly highlighted. This contribution lies in integrating techniques in four critical points, namely for simulation planning, base simulation validation, case-study experimentation and result analysis.  The decisions have also been well justified.

However, I would like to point out the following:

In section 4.1, authors state that they chose to work with a limited group of 10 terrestrial and homogeneous mobile robots, executing decentralized rendezvous task. I am wondering, since this has been done in a simulation, why limiting the number of robots to just 10? What is the impact of the number of robots on (1) the results of the MPC algorithm, (2)topology control and (3) the convergence time considering techniques with message losses and without message losses?

Line 349…The method second step is… This needs to be corrected.

Reviewer 2 Report

This manuscript proposes an e-Robotic framework to simulate networked multi-robot systems (MRS), which can be seen as Networked Control Systems (NCS). The specificity of the proposed framework denoted IRoNS (Integrated Robotic and Network Simulation) is that it models and takes into accoount the important effects of the network limitations on the effectiveness of the MRS while operating in real life. The authors claim that going from simulated environments to real-life testbeds is a challenge because of network issues, and they are absolutely right.

The research question is clearly framed and the problem tackled in this manuscript is timely as the field of MRS is rapidly expanding. Moreover, the paper is well written. Having said that, there are a number of minor points that should be addressed by the authors before this manuscript can be accepted. These comments/questions are listed below:
(1) When one considers the limitations of the interconnecting network, the following aspects have to be accounted for: (i) limited bandwidth, (ii) time delays, and (iii) packet dropouts. It is not very clear to me whether the authors are considering all these real practical challenges as part of their IRoNS framework.
(2) The authors clearly delineate (i) Problem definition, (ii) Simulation framework, and (iii) Experimentation. The central weakness of the present work lies with (iii) Experimentation, and the lack of ``real'' experimentation with actual robots. Unfortunately, it is impossible to truly assess the real effectivneness of the proposed framework.
(3) Another problem is that, from reading the paper, one gets the impression that the issues related to the network are purely technical or practical ones associated with the inherent limitations of any network topology. However, this is fundamentally wrong and the authors must correct this important misconception. For instance, it has been shown recently that the topology of the interconnecting network greatly affects the overall dynamics of a multi-agent system:
[1] Mateo et al. Effect of correlations in swarms on collective response. Scientific reports, 7(1), p.10388, 2017.
and also with multi-robot systems:
[2] Mateo et al. Optimal network topology for responsive collective behavior. Science Advances, 5(4), p.eaau0999, 2019.
Actually the framework proposed by the authors in this manuscript could help experimentations such as those reported in [2]. The authors must highlight these references and correct this unintended conceptual error.
(4) I very much appreciated the up-to-date literature review. However, since the authors consider a platform-agnostic and network-agnostic framework, they should cite on page 2, in the paragraph (lines 48-54) a similar platform agnostic framework reported in Ref. [3] below. However, in Ref. [3], the framework also includes the possibility to directly embed the simulation framework within the MRS.
[3] Chamanbaz et al. Swarm-enabling technology for multi-robot systems. Frontiers in Robotics and AI, 4, p.12, 2017.
(5) Regarding Section 4.2.2: what kind of network mdodules are actually considered, nRF32? Zigbee?
(6) Section 4.2.5 Topology Control: I have been quite puzzled by this section, in which the authors decide to enforce a static topology to avoid the challenges of a switching topology. My surprise comes from the fact that it is well knwon that this switching in the network are actually helping the system in its collective dynamics (see Refs. [1] & [2] above).
